# A simulation framework to determine optimal strength training and musculoskeletal geometry for sprinting and distance running

**Tom Van Wouwe**[1]*, **Jennifer Hicks**[1], **Scott Delp**[1], **Karen C. Liu**[2]

**1** Department of Bioengineering, Stanford University, Stanford, California, United States of America,
**2** Department of Computer Science, Stanford University, Stanford, California, United States of America

* tvwouwe@stanford.edu

**Data Availability Statement:** All code is available at: https://github.com/tomvanwouwe1992/MusculoskeletalSpecialization.

## Abstract

Musculoskeletal geometry and muscle volumes vary widely in the population and are intricately linked to the performance of tasks ranging from walking and running to jumping and sprinting. As an alternative to experimental approaches, where it is difficult to isolate factors and establish causal relationships, simulations can be used to independently vary musculoskeletal geometry and muscle volumes, and develop a fundamental understanding. However, our ability to understand how these parameters affect task performance has been limited due to the high computational cost of modelling the necessary complexity of the musculoskeletal system and solving the requisite multi-dimensional optimization problem. For example, sprinting and running are fundamental to many forms of sport, but past research on the relationships between musculoskeletal geometry, muscle volumes, and running performance has been limited to observational studies, which have not established cause-effect relationships, and simulation studies with simplified representations of musculoskeletal geometry. In this study, we developed a novel musculoskeletal simulator that is differentiable with respect to musculoskeletal geometry and muscle volumes. This simulator enabled us to find the optimal body segment dimensions and optimal distribution of added muscle volume for sprinting and marathon running. Our simulation results replicate experimental observations, such as increased muscle mass in sprinters, as well as a mass in the lower end of the healthy BMI range and a higher leg-length-to-height ratio in marathon runners. The simulations also reveal new relationships, for example showing that hip musculature is vital to both sprinting and marathon running. We found hip flexor and extensor moment arms were maximized to optimize sprint and marathon running performance, and hip muscles the main target when we simulated strength training for sprinters. Our simulation results provide insight to inspire future studies to examine optimal strength training. Our simulator can be extended to other athletic tasks, such as jumping, or to non-athletic applications, such as designing interventions to improve mobility in older adults or individuals with movement disorders.

**Funding:** TVW was supported to perform this work by the Joe and Clara Wu Tsai Foundation through the Wu Tsai Human Performance Alliance. The funders had no role in study design, data collection and analysis, decision to publish, or preparation of the manuscript.

**Competing interests:** The authors declare that they have no conflict of interest.

## Author summary

Our study addresses the challenge of determining optimal musculoskeletal parameters for tasks like sprinting and marathon running. Existing research has been limited to observational studies and simplified simulations. To overcome these limitations, we developed a differentiable musculoskeletal simulator to optimize running performance. We replicated past findings and uncovered new insights. We confirmed the benefits of increased muscle mass for sprinters and identified key factors for marathon runners, such a mass in the lower end of the healthy BMI range and an increased leg-length-to-height ratio. Hip musculature was found to be critical for both sprinting and marathon running. Our simulation results have practical implications. They can inform customized strength training for sprinters and marathon runners. Additionally, the simulator can be extended to other athletic tasks, benefiting various sporting events. Beyond athletics, our open-source simulator has broader applications. It can determine minimal strength requirements for daily activities, guide strength training in the elderly, and estimate the effects of simulated musculoskeletal surgery.

## Introduction

Performance of tasks ranging from rising from a chair to competing in Olympic-level sporting events depends on precise coordination of many muscles. Variations in an individual's musculoskeletal geometry and muscle volumes affect performance for many movement tasks [1–3], but our ability to identify cause-effect relationships has been limited because quantifying the effects of changing a person's musculoskeletal geometry or muscle volumes on task performance is complex.

Musculoskeletal simulation could allow researchers to quantify the effects of variations in body segment dimensions and muscle properties on performance. Simulation allows us to observe the influence of variables that cannot be changed in an experiment (e.g., the height of a runner) and enables us to examine whether specific muscular and skeletal features that have been associated with the performance of a task causally contribute to task performance. Previous simulation studies have opened the door to this possibility but have been limited by simplified representations of the musculoskeletal system [4–6]. The majority use gradient-based trajectory optimization, which requires gradient computations that are computationally expensive because there are typically non-differentiable expressions requiring the use of finite differentiation [7]. To understand the effects of body segment dimensions and muscle properties on task performance using finite differentiation to compute gradients will lead to days or weeks of computation time. Automatic differentiation, a faster alternative to finite differences, has been applied to speed up musculoskeletal simulation (e.g. [8,9]). More recently, Falisse et al. [10] implemented a framework that is based on the commonly used OpenSim models and software [11]. However, the algorithm requires body segment dimensions and muscle properties to be fixed. Our simulator can optimize both body segment dimensions and muscle volumes of a complex, three-dimensional muscle-driven musculoskeletal model for a range of tasks. By implementing a musculoskeletal simulator that is fully differentiable with respect to both body segment dimensions and muscle volumes, we have enabled simulations to be completed within hours on a standard computer. To test and apply this new simulator, we used it to study the role of body segment dimensions and muscle volumes in sprinting and marathon running performance.

The 100m dash is arguably the most prestigious event in track-and-field [12,13], and marathon running serves as a frontier of human endurance [14,15]. Success in these two events has been associated with different skeletal and muscular features, largely through observational studies. For example, Sedeaud et al. [1] found that mean height, mean body mass index (BMI), and variability in BMI decreased with increasing distance of the event in which male runners specialized. Size, proportions, and other aspects of musculoskeletal geometry have also been associated with an advantage in sports such as speed skating and swimming [2], and cycling [3].

Compared to the general population, sprinters exhibit specialization in musculoskeletal geometry, such as a more limited height range (i.e., sprinters are typically not very short or tall) [16]. Although extremes of height are rare in sprinters [16], body height was not associated with performance in 100m personal best times in a group of sprinters [17]. Body proportions, however, might be important for sprint performance. For example, Tomita et al. [18] found that a greater tibia-to-femur length was associated with better performance in 400m runners. They posit that a greater tibia-to-femur length reduces the leg's moment of inertia with respect to the hip and thus reduces positive work done by the hip flexors during the swing phase of running. Top sprinters have highly developed musculature for both the lower and upper body, as established via both medical imaging [19] and external measurements [17,20–22]. In a group of competitive sprinters, greater thigh girth, hip strength, and body mass index (BMI) was associated with greater sprint performance as assessed via personal best times [17,23]. Further, greater relative muscle volume in the hip muscles has been shown to differentiate elite sprinters, sub-elite sprinters, and non-sprinters [19]. With hip musculature suggested as a limiter of sprinting performance, stronger hip muscles and a deeper (i.e. greater along the anteroposterior axis) pelvis, which increases the moment arms of hip muscles, might be advantageous.

Compared to the general population, distance runners are shorter, have lower BMIs [20,24] and have a greater tibia-to-femur length ratio [25,26]. In trained distance runners, a greater relative tibia-to-femur length ratio is also associated with better running performance [26], as is a greater relative lower limb length [25,27]. Distance runners, compared to sprinters, have lower maximal isometric knee flexor and extensor torques, in absolute terms and when normalized by body weight [28].

While the observational studies described above reveal associations between performance and musculoskeletal features, they are limited in their ability to identify cause-effect relationships. To overcome this, Deane et al. [29] conducted an interventional experiment, which revealed that hip flexor training significantly improves 40-yd dash times. However, such interventional studies are rare, costly, and difficult to control (e.g., hip flexor training might increase strength of additional muscle groups).

Past simulation studies have explored causal relationships between a musculoskeletal model's capacity (e.g., maximal joint torques or muscle force) and performance in sprinting [30], jumping [4,5,31] and gymnastics [32]. However, these studies have been limited in their ability to capture the complexity of the musculoskeletal system as it applies to athletic performance [33]. For example, previous simulation studies that investigated jumping performance did not include muscles [5] or included only six muscles and were restricted to two dimensions [4,31]. Two simulation studies have revealed how muscle force-length-velocity relationships influence sprinting capacity, but were limited to 2D models driven by nine muscles [6,34]. No simulation studies have comprehensively investigated how differences in body-segment dimensions affect running performance.

In this study, we developed a three-dimensional musculoskeletal simulator that is fully differentiable with respect to both body-segment dimensions and muscle properties to analyze

the effects of body-segment dimensions, muscle volume, and distribution of muscle volume on sprinting and marathon running performance. We sought to determine how body-segment dimensions affect maximal sprinting speed and the metabolic cost of running a marathon at moderate speed. We also simulated how optimized, targeted strength training improves sprinting and marathon running performance. Understanding the influence of body segment dimensions and muscle volume distribution on performance can inform the selection of sports in which to compete and personalize training to help maximize performance. Our simulator is open source and enables researchers to conduct additional studies investigating the relationships between musculoskeletal parameters and increased performance, as in sport, or reduced performance, as can result from injuries, diseases, and disorders.

## Methods

### Overview of performed simulations

We performed ten predictive simulations of a running gait (overview in Table 1) using the three-dimensional musculoskeletal model developed by Hamner and colleagues [32], which includes 92 muscle actuators, 8 torque actuators, and 31 degrees of freedom. The generic model represents a male subject with a stature of 1.81m and mass of 75kg, corresponding to the 73$^{rd}$ and 63$^{rd}$ percentiles for 30 year old males in the United States [35], and that matches the strength of a young healthy subject capable to performing athletic activities. The model has been tested and used in many previous studies, including studies of walking [10], running [10,36] and sprinting [37,38].

To simulate marathon running, we imposed a running speed of 3.33m/s and minimized lower limb energy expended over the marathon distance [10]. To simulate sprinting, running speed was maximized for a single simulated half gait cycle. We thus ignore the acceleration phase of sprinting and limit ourselves to simulating the maximal velocity. In all simulations, we optimized the excitations of the muscles and torque actuators to achieve the performance criterion of interest (i.e., marathon or sprint performance).

To answer our research questions, we performed simulations with a generic model that maximize sprinting performance (sim1) and marathon performance (sim2). Next, we

**Table 1. Summary of the ten predictive simulations.** The task simulated is either sprinting or marathon running. Motor coordination is always a decision variable, whereas body segment factors and muscle volume scaling factors are either set to the values of a model or also optimized. The model name describes which model is used or results from the simulation. We finally mention which figure in the results section displays results from each simulation.

| | Task | Motor coordination | Body segment scaling factor | Muscle volume scaling factor | Model name | Figure |
|---|---|---|---|---|---|---|
| 1 | Sprinting | decision variable | generic | generic | generic | 1.A,1.C,1.D,2.A |
| 2 | Marathon | decision variable | generic | generic | generic | 1.B,1.C,1.D,2.B |
| 3 | Sprinting | decision variable | decision variable | scaled with body mass | sprint-optimized body segments | 1.A,1.C,1.D,1.E,1.F |
| 4 | Marathon | decision variable | decision variable | scaled with body mass | marathon-optimized body segments | 1.B,1.C,1.D,1.E,1.F |
| 5 | Marathon | decision variable | sprint-optimized body segments | sprint-optimized body segments | sprint-optimized body segments | 1.B |
| 6 | Sprinting | decision variable | marathon-optimized body segments | sprint-optimized body segments | marathon-optimized body segments | 1.A |
| 7 | Sprinting | decision variable | generic | decision variable | sprint-optimized muscle volumes | 2.A, 2.C, 2.D, 3 |
| 8 | Marathon | decision variable | generic | decision variable | marathon-optimized muscle volumes | 2.B, 2.C, 2.D, |
| 9 | Marathon | decision variable | generic | sprint-optimized muscle volumes | sprint-optimized muscle volumes | 2.B |
| 10 | Sprinting | decision variable | generic | marathon-optimized muscle volumes | marathon-optimized muscle volumes | 2.A, 3 |

performend simulations that optimized body-segment dimensions (i.e., the three-dimensional dimensions of each segment) for sprinting (sim3) and marathon performance (sim4). When scaling the skeleton, we chose to scale muscle volumes proportionally to the change in whole body mass with a 1:1 ratio. Individual body segments are scaled by assuming constant density. As such, the mass percentage of muscle and fat tissue remains constant for models of different sizes and thus a heavier model has stronger muscles. Scaling the segments affects the bone geometry, muscle attachment points, muscle moment arms, and muscle properties, such as optimal fiber length and tendon slack length. As such, these simulations yielded a model with sprint-optimized skeleton body segment dimensions and a model with marathon-optimized body segment dimensions. Next, we evaluated these new models on the opposite tasks being marathon running (sim5) and sprinting respectively (sim6). Next, we performed simulations that optimized the distribution of added muscle volume (i.e., optimal distribution of muscle volume when adding 5% of the total muscle volume) for sprinting (sim7) and marathon running (sim8) which yielded models with sprint-optimized and marathon-optimized muscle volumes. Finally, we also evaluated these models on the opposite task (sim9 and sim10).

### Differentiable musculoskeletal simulator

We developed a differentiable musculoskeletal simulator (Fig 1) to enable efficiently performing the described simulations. Every function within this simulator is differentiable; thus, we can rapidly obtain the gradient of the output with respect to all its inputs using automatic differentiation rather than finite differencing, and all gradients are continuous.

At each timestep of a simulation our musculoskeletal simulator gives the state derivatives given the state and decision variables. The state $\left(\boldsymbol{x} \in \mathbb{R}^{254}\right)$ of the musculoskeletal simulator is determined by the activations $\left(\boldsymbol{a}_m \in \mathbb{R}^{92}\right)$ of the 92 included lower limb muscles, the force in each tendon $\left(\boldsymbol{F}_t \in \mathbb{R}^{92}\right)$, the activation of the eight torque actuators of the upper limb degrees of freedom $\left(\boldsymbol{a}_T \in \mathbb{R}^{8}\right)$, and the generalized positions $\left(\boldsymbol{q} \in \mathbb{R}^{31}\right)$ and velocities $\left(\dot{\boldsymbol{q}} \in \mathbb{R}^{31}\right)$, including the six degrees-of-freedom of the pelvis and 25 joint angles:

$$\boldsymbol{x} = [\boldsymbol{a}_m; \boldsymbol{a}_T; \boldsymbol{F}_t; \boldsymbol{q}; \dot{\boldsymbol{q}}]$$

The decision variables that form the input to the musculoskeletal simulator at each timestep are the muscle $\left(\boldsymbol{e}_m \in \mathbb{R}^{92}\right)$ and torque actuator excitations $\left(\boldsymbol{e}_T \in \mathbb{R}^{8}\right)$, the scaling factors of the skeleton segments $\left(\boldsymbol{p}_s \in \mathbb{R}^{3 \times 18}\right)$ and scaling factors of the muscle volumes $\left(\boldsymbol{p}_{V_{muscle}} \in \mathbb{R}^{92}\right)$.

The state derivatives are described by muscle activation dynamics [40], with activation and deactivation time constants 15ms and 60ms respectively:

$$\dot{\boldsymbol{a}}_m = \boldsymbol{f}_{act,musc}(\boldsymbol{e}_m, \boldsymbol{a}_m),$$

muscle-tendon dynamics [40]:

$$\dot{\boldsymbol{F}}_t = \boldsymbol{f}_{mt}\left(\boldsymbol{a}, \boldsymbol{F}_t, \boldsymbol{l}_{mt}(\boldsymbol{q}, \boldsymbol{p}_s), \dot{\boldsymbol{l}}_{mt}(\boldsymbol{q}, \dot{\boldsymbol{q}}, \boldsymbol{p}_s), \boldsymbol{p}_m(\boldsymbol{p}_s)\right),$$

torque actuator activation dynamics [10]:

$$\dot{\boldsymbol{a}}_T = \boldsymbol{f}_{act,torque}(\boldsymbol{e}_T, \boldsymbol{a}_T) = \frac{\boldsymbol{e}_T - \boldsymbol{a}_T}{0.035},$$

and skeleton dynamics [41]:

$$\ddot{\boldsymbol{q}} = \boldsymbol{f}_s(\boldsymbol{q}, \dot{\boldsymbol{q}}, \boldsymbol{F}_t, \boldsymbol{p}_s) = M(\boldsymbol{q}, \boldsymbol{p}_s)^{-1}\left[G(\boldsymbol{q}, \boldsymbol{p}_s) + C(\boldsymbol{q}, \dot{\boldsymbol{q}}, \boldsymbol{p}_s) + \boldsymbol{\tau}(\boldsymbol{F}_t, \boldsymbol{a}_T, \boldsymbol{q}, \dot{\boldsymbol{q}}, \boldsymbol{p}_s) + \boldsymbol{f}_c(\boldsymbol{q}, \dot{\boldsymbol{q}}, \boldsymbol{p}_s)\right],$$

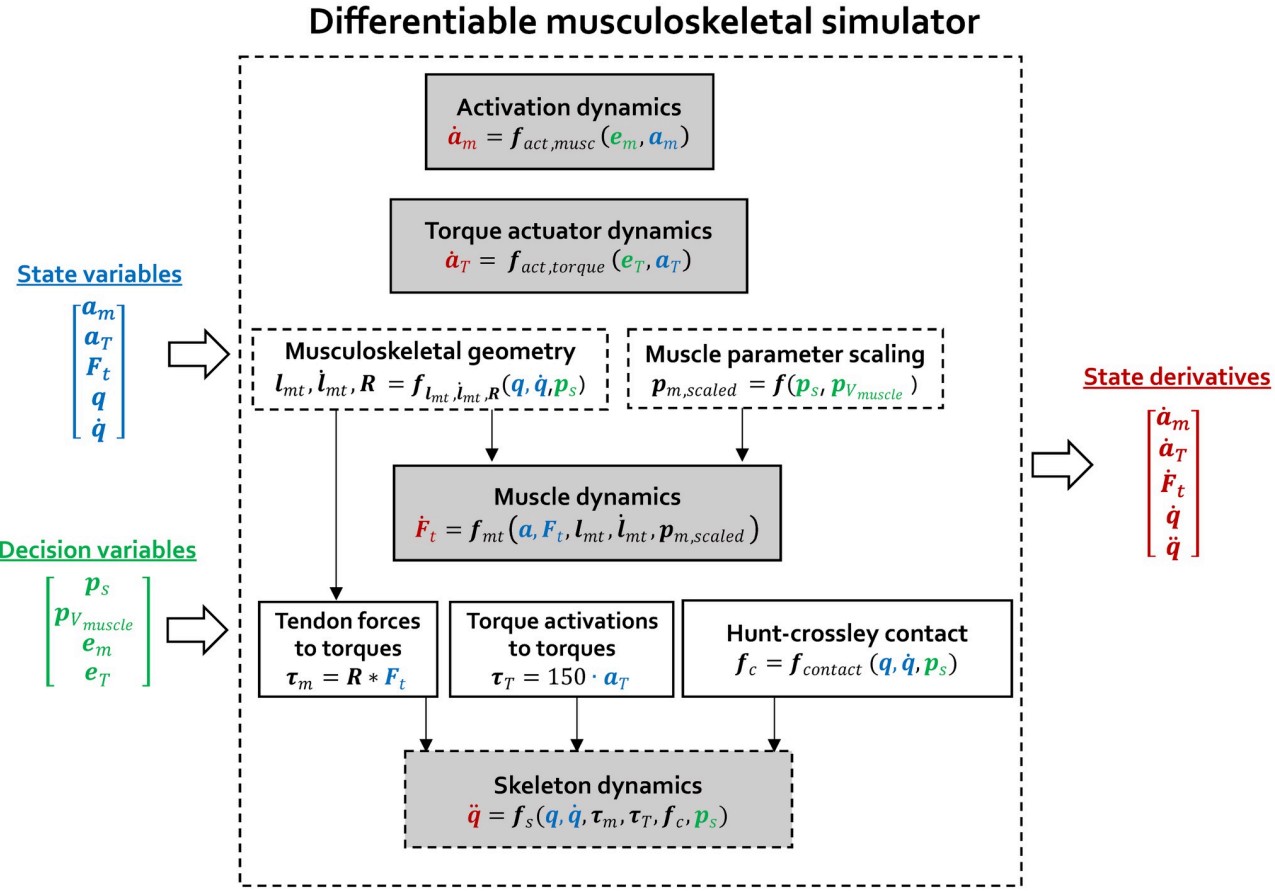

**Fig 1.** Our differentiable musculoskeletal simulator generates the derivatives of the state variables given the state variables (muscle activations $a_m$, torque actuator activations $a_T$, tendon forces $F_t$, generalized positions $q$ and velocities $\dot{q}$) and the decision variables (skeleton segment scaling factors $p_s$, muscle volume scaling factors $p_{V_{muscle}}$, muscle excitations $e_m$, torque actuator excitations $e_T$). This is achieved by evaluating a set of dynamics equations: activation dynamics, torque actuator dynamics, muscle dynamics, and skeleton dynamics. Evaluating muscle and skeleton dynamics depends on the outputs of musculoskeletal geometry computations (i.e., muscle-tendon lengths $l_{mt}$ and velocities $\dot{l}_{mt}$ and muscle moment-arm matrices $R$) and on the scaled muscle parameters ($p_{m,scaled}$). Since the scaling of the skeleton and muscle volumes are decision variables, we formulated musculoskeletal geometry computation, muscle parameter scaling and skeleton dynamics as a differentiable function of these decision variables. The dotted boxes indicate the parts of the simulator where we turned non-differentiable computation used in OpenSim and Falisse et al. [39] into differentiable computation. Tendon forces are mapped to joint muscle torques ($\tau_m$) by the moment-arm matrix ($R$). Torque actuator activations are scaled to torque actuator torques ($\tau_T$) by a scaling factor of 150 [10]. A contact function ($f_{contact}$) based on the Hunt-Crossley contact model gives the generalized forces resulting from contact ($f_c$).

where $e_m$ are the muscle excitations, $l_{mt}$ muscle-tendon lengths, $\dot{l}_{mt}$ muscle-tendon velocities, $p_s$ the skeleton parameters, $p_m$ the muscle parameters, $e_T$ the torque actuator excitations, $\tau$ biological joint torques, $f_c$ the function describing the generalized forces that result from contact, $M$ the mass matrix, $G$ the vector of gravitational forces, and $C$ the vector of Coriolis and centrifugal forces. The force-length and force-velocity relationships are identical to [40] and were not adapted to reflect specialized characteristics due to different fiber type distributions in sprinters vs distance runners [42,43]. Contact is modelled using a Hunt-Crossley model and occurred between eight contact spheres attached to each of the feet and the ground. The location and properties of these contact spheres are as described in [39].

The biological joint torques are the result of the torques generated by the muscles ($\tau_m$), the torques generated by the torque actuators for the upper limbs ($\tau_T$) and passive joint torques

$(\boldsymbol{\tau}_{pas})$:

$$\boldsymbol{\tau} = \boldsymbol{\tau}_m + \boldsymbol{\tau}_T + \boldsymbol{\tau}_{pas},$$

$$\boldsymbol{\tau}_m = R(\boldsymbol{q}, \boldsymbol{p}_s){*}\boldsymbol{F}_t,$$

$$\boldsymbol{\tau}_T = 150{*}\boldsymbol{a}_T,$$

$$\boldsymbol{\tau}_{pas} = \boldsymbol{k}_{U,1}.e^{\boldsymbol{k}_{U,2}(\boldsymbol{q}-\boldsymbol{\theta}_U)} + \boldsymbol{k}_{L,1}.e^{\boldsymbol{k}_{L,2}(\boldsymbol{\theta}_L-\boldsymbol{q})} + 0.1\dot{\boldsymbol{q}}$$

The muscle torques result from matrix multiplication between the muscle forces with $R(\boldsymbol{q}, \boldsymbol{p}_s)$ the 92x31 matrix of moment arms of the muscles with respect to the joints. The torque actuator torques result from scaling the torque actuator activations that are bounded between -1 and 1 with 150Nm as a scaling factor mapping torque actuator activation to torque [10,38].

The passive joint torques ($\boldsymbol{\tau}_{pas}$) consist of joint limit torques to model ligaments for the muscle driven joints and damper joint torques for all joints with a damping constant of 0.1 Nm.s/rad. The joint limit torques are parametrized by six parameters ($\boldsymbol{k}_{U,1}$, $\boldsymbol{k}_{U,2}$, $\boldsymbol{\theta}_U$, $\boldsymbol{k}_{L,1}$, $\boldsymbol{k}_{L,2}$, $\boldsymbol{\theta}_L$) describing the exponential decay and increase across the range of motion for every joint coordinate. The parameters are taken from [44].

Finally, for the metatarsophalangeal joint, which is not muscle actuated, a spring-damper joint torque is added that has different parameters depending on whether sprinting (stiffness: 40 Nm/rad, damping: 0.4 Nm.s/rad) [38] or marathon running (stiffness: 25 Nm/rad, damping: 1.9 Nm.s/rad) [39] is simulated.

The skeleton parameters $\boldsymbol{p}_s \in \mathbb{R}^{3x18}$ consist of three scaling factors for each body segment, one for scaling each dimension. When scaling the model, we assume the following 18 bodies: the pelvis, trunk+head, plus two of each of the talus, mid- and hindfoot, toes, shank, thigh, upper arms, lower arms, hands. To simplify the problem, we use constraints to impose symmetry and to require identical scaling factors for the trunk+head, upper arms, lower arms, and hands, and identical scaling factors for talus, mid- and hindfoot and toes. Importantly, scaling the body segments affects the following quantities in the skeleton dynamics: $M$, $G$, $C$, $\boldsymbol{f}_c$, $\boldsymbol{\tau}$ defined above.

The skeleton segment scaling factors are also an input to the musculoskeletal geometry computation that calculates the muscle-tendon lengths, muscle-tendon velocities and moment-arm matrix as a function of the generalized coordinates, generalized velocities and skeleton segment scaling factors:

$$\left[\boldsymbol{l}_{mt}, \dot{\boldsymbol{l}}_{mt}, R\right] = \boldsymbol{f}_{l_{mt}, \dot{l}_{mt}, R}(\boldsymbol{q}, \dot{\boldsymbol{q}}, \boldsymbol{p}_s).$$

The Hill-type muscle parameters $\boldsymbol{p}_m$ consist of: the muscle physiological cross sectional area (**PCSA**), the specific tension of muscle fibers ($\sigma$), tendon slack length ($\boldsymbol{l}_{T,s}$), optimal fiber length ($\boldsymbol{l}_{m,opt}$), pennation angle ($\boldsymbol{\alpha}_m$) and tendon stiffness ($\boldsymbol{k}_T$) [40]. The cross-sectional area and specific tension determines the muscle maximal isometric force:

$$F_{max,iso} = PCSA{*}\sigma$$

From the PCSA and the optimal fiber length, the muscle volume is calculated:

$$V_{muscle} = PCSA{*}\boldsymbol{l}_{m,opt}$$

which is an input to several other computations including the metabolic energy consumption [45].

To mimic strength training, we allowed $V_{muscle}$ of individual muscles to be scaled by a scaling factor $\boldsymbol{p}_{V_{muscle}}$:

$$V_{muscle,scaled} = \boldsymbol{p}_{V_{muscle}} \cdot V_{muscle}$$

and as such changing the muscle maximal isometric force. When the skeleton was scaled $V_{muscle}$ changed with a 1:1 ratio to the change in skeleton mass. In this case, PCSA could not be directly calculated from $V_{muscle,scaled}$ as $l_{m,opt}$ was first scaled to $l_{m,opt,scaled}$ (see next paragraph).

When scaling the skeleton, the tendon slack length and optimal fiber length are adapted as well depending on the total length change of the muscle tendon unit length when the model is placed in the anatomical pose ($\boldsymbol{q} = \boldsymbol{0}$):

$$\boldsymbol{l}_{m,opt,scaled} = \boldsymbol{l}_{m,opt} \cdot \frac{\boldsymbol{f}_{l_{mt}}(\boldsymbol{0}, \boldsymbol{p}_s)}{\boldsymbol{f}_{l_{mt}}(\boldsymbol{0}, \boldsymbol{1})}.$$

The same ratio is applied to scale tendon slack length.

Scaled muscle parameters are thus a function of original muscle parameters, the scaling factors of the skeleton segments and the scaling factor for the muscle volume:

$$\boldsymbol{p}_{m,scaled} = \boldsymbol{f}(\boldsymbol{p}_m, \boldsymbol{p}_s, \boldsymbol{p}_{V_{muscle}})$$

## Turning non-differentiable into differentiable computation

Optimizing the skeleton body segment dimensions is not feasible using the state-of-the-art musculoskeletal simulator OpenSim [11] and Simbody [41] due to its non-differentiable computation. The simulator from Falisse et al. [10] enables differentiable computation for a part of the OpenSim-Simbody simulator and serves as a starting point for our simulator. However, the simulator of from Falisse et al. [10] is not differentiable with respect to all musculoskeletal variables of interest. As such, an important technical contribution of our work is to make the entire musculoskeletal simulator differentiable. We identified two non-differentiable operations when relying on OpenSim and Simbody. First, musculoskeletal geometry computations, $l_{mt}, \dot{l}_{mt}, R = f_{l_{mt}, \dot{l}_{mt}, R}(\boldsymbol{q}, \dot{\boldsymbol{q}}, \boldsymbol{p}_s)$, are non-differentiable with respect to generalized coordinates ($\boldsymbol{q}$), velocities ($\dot{\boldsymbol{q}}$) and skeleton segment scaling factors ($\boldsymbol{p}_s$), and the first-order derivatives are not guaranteed to be continuous. Second, the skeleton dynamics, $\ddot{\boldsymbol{q}} = \boldsymbol{f}_s(\boldsymbol{q}, \dot{\boldsymbol{q}}, \boldsymbol{F}_t, \boldsymbol{p}_s)$, are non-differentiable with respect to $\boldsymbol{p}_s$, which determines the contributions of $M$, $G$, $C$, $\boldsymbol{f}_c$ to the skeleton dynamics.

## Differentiable musculoskeletal geometry computation

We implemented the musculoskeletal geometry computation as a differentiable neural network function:

$$l_{mt}, \dot{l}_{mt}, R = f_{l_{mt}, \dot{l}_{mt}, R}(\boldsymbol{q}, \dot{\boldsymbol{q}}, \boldsymbol{p}_s).$$

Musculoskeletal geometry computation in OpenSim is executed as follows: first, based on the skeleton segment scaling factors the bone geometries, muscle attachment points, muscle via points and muscle wrapping surfaces are adapted, next, using the scaled geometry the muscle-tendon lengths and moment arms are calculated. Both parts are implemented in OpenSim as non-differentiable operations with non-continuous first-order derivatives.

To resolve this, we implemented a shallow (two hidden layers) neural network to calculate $l_{mt}$, $\boldsymbol{R}$ for every muscle. Having a differentiable function for the computation of $l_{mt}$ for each pose ($\boldsymbol{q}$) and set of skeleton segment scaling factors ($\boldsymbol{p}_s$) allows us to have a differentiable function to perform muscle parameter scaling $\boldsymbol{p}_{m,scaled} = \boldsymbol{f}(\boldsymbol{p}_m, \boldsymbol{p}_s, \boldsymbol{p}_{V_{muscle}})$. Finally, the muscle-tendon velocities are computed using the chain rule:

$$\dot{\boldsymbol{l}}_{mt} = \frac{\partial \boldsymbol{l}_{mt}}{\partial \boldsymbol{q}} \cdot \dot{\boldsymbol{q}}.$$

Having a separate neural network for every muscle reduces computational complexity, as we can take into account that every muscle only attaches to a limited number of bones.

We used OpenSim to generate training data and started from the generic OpenSim model for running [36]. We scaled 2,000 versions of this model using the OpenSim Scale tool [11]. The scaling factors, that serve as input to the Scale Tool, are a 54x1 vector (representing the three dimensions of the 18 bodies of the model), of which each element is drawn from a uniform distribution: $U(0.8,1.2)$. Based on variation in the ANSUR II dataset, we chose the respective lower and upper limits of 0.8 and 1.2 as these cover most of the variation in an adult population [46].

With the scaled models at our disposal, we generated training samples for all the different muscles. For every training sample, we randomly selected one of the 2000 models and randomly drew a joint pose from a uniform distribution that has lower and upper limits according to the joint range of motion. We put the model in that joint pose and observed the muscle tendon length and moment arm with respect to the joint coordinates that are actuated by the muscle. Depending on whether the muscle actuates, one, two or more joint coordinates, we drew 20,000, 82,000, or 200,000 training samples for that muscle.

We then trained a separate neural network for each muscle using Adam [47], a stochastic gradient descent algorithm commonly used to optimize neural networks, with a mean-squared-error loss over 1,000 epochs and a batch size of 64. We used feedforward neural networks with 'tanh' activation functions and two hidden layers. The size of the hidden layers was 8, 12, or 16 depending on whether the muscle actuates 1, 2 or more joint coordinates.

We experimented to minimize the size of the neural networks to reduce computational complexity without sacrificing accuracy of the approximation. With the described set-up we confirmed that our predictive simulations of running at 3.33m/s yielded kinematics and muscle activation very close to those with the original OpenSim model and for OpenSim models with all scaling factors at 0.85 or 1.15.

The neural networks approximate the relationships in OpenSim between (1) joint coordinates and segment dimensions and (2) moment arms and muscles lengths. Discontinuities and non-smooth transitions in OpenSim moment arms and muscle lengths are smoothed because of the relatively shallow and small neural networks we used.

## Differentiable skeleton dynamics

Skeleton dynamics are based on SimBody [41]. We adapted the source code transformation tool from [48] to enable automatic differentiation of the skeleton dynamics with respect to the segment scaling parameters $\boldsymbol{p}_s$. This source code transformation tool analyzes a given function's source code and outputs the gradient of that function. The source code transformation tool takes a customized.cpp description of the SimBody skeleton model and its skeleton dynamics as a function to analyze and differentiate. The tool also requires a description of the variables with respect to which it will generate the gradient. In addition to the state-of-the-art implementation, where the functions take generalized coordinates, velocities, and

accelerations as differentiable input, we extended its functionality to be differentiable with respect to the geometrical scaling factors of each segment. Therefore, we added functions to define how these geometrical segment scaling factors change the mass, center-of-mass location, and inertial properties for every segment. Next, we added functions to define how these segment scaling factors change rotational and translational offsets across the kinematic tree of the skeleton. These functions mimicked how segment scaling factors affect these model properties when using the OpenSim Scale tool, but in a smooth and differentiable way.

## Trajectory optimization

Each predictive simulation was solved as a trajectory optimization problem. We simulated steady-state gait (running and sprinting) and assumed symmetry. As such we only needed to simulate half a gait cycle while imposing symmetry, as well as continuity and periodicity constraints for the appropriate states.

For every trajectory optimization problem, we optimized at least the motor coordination, consisting of muscle ($e_m$) and torque excitations ($e_T$), and the initial state of the musculoskeletal system ($x(t_0)$).

For simulations where we optimized either body segment dimensions or muscle volume scaling to maximize task performance, we solved a trajectory optimization problem where either the skeleton scaling parameters ($p_s$) or the scaling of muscle volumes ($p_{V_{muscle}}$) were added to the optimization variables. Note that if $p_s$ is optimized this directly determines $p_{V_{muscle}}$ as muscle volume is scaled with change in mass.

## Objective function

The minimal energy objective for marathon running was adapted from Falisse et al. [10] and consists of five main contributions:

$$J_{marathon} = \frac{J_{metabolic} + J_{a_m} + J_{acc} + J_{limit} + J_{a_T}}{l_{gait\ cycle}}$$

with $J_{metabolic}$ the squared muscle metabolic energy based on the Bhargava model [45] of metabolic energy expenditure, $J_{a_m}$ the sum of squared muscle activations modelling muscle fatigue, $J_{acc}$ the sum of squared joint accelerations modelling motion smoothness, $J_{limit}$ the sum of squared limit joint torques modelling avoidance of ligament strain, and $J_{a_T}$ the sum of squared upper limb torque actuator activations modelling upper body fatigue and energy expenditure. The sum of these terms is normalized by the length of the gait cycle. The metabolic cost of running a marathon, which is reported in the results as marathon performance is computed by multiplying $\frac{J_{metabolic}}{l_{gait\ cycle}}$ with the length of a marathon: 42,196 m.

A straightforward sprint objective to maximize the average velocity was chosen:

$$J_{sprint} = -v_{avg}^2 + 1e^{-6} * J_{marathon}$$

with a small contribution of the marathon energy term to improve the numerical condition of the optimization problem.

## Constraints and bounds

Muscle excitations and activations are bound to be between 0 and 1, whereas torque actuator excitations are bounded between -1 and 1. We include path constraints to avoid penetration between body segments and additional bounds for the joint ranges of motion. These ranges of

motion are generous and typically not reached for most degrees of freedom as the modelled passive forces representing ligaments provide a physical joint limit. However, for the upper limb degrees of freedom we use a more strict and representative range of motion that was reached in some simulations. This choice was made as we did not model muscular or ligamentous structures to limit these. Similarly, hip inward (-10˚) and outward rotation (+10˚) as well as knee extension (0˚) bounds were typically reached. For the remaining variables (muscle lengths, joint velocities, muscle velocities, joint accelerations) we use generous bounds that were there to improve numerical stability during optimization and were not reached.

For the simulations of marathon running we imposed the average speed to be 3.33m/s.

For simulations where the skeleton scaling parameters, $p_s$, were optimized, these were bounded between 0.8 and 1.2. We also imposed a constraint on the Body Mass Index to be between 17.5 and 25.5 to represent a healthy person.

When simulating strength training, the increase of individual muscle volumes was limited to 20% and the total increase in muscle volume summed over all muscles was limited to 5% of the initial total muscle volume.

### Direct collocation and implicit dynamics

To improve numerical conditioning, we formulated muscle and skeleton dynamics with implicit rather than explicit differential equations. We therefore introduced derivatives of tendon force and coordinate accelerations as additional controls, and we imposed the nonlinear dynamic equations describing muscle contraction and skeleton dynamics as algebraic constraints.

We used direct collocation to transcribe each trajectory optimization problem into a large sparse nonlinear program. We used a third-order Radau quadrature collocation scheme with 50 mesh intervals per half gait cycle and solved the resulting NLP with the solver IPOPT. All gradients were computed using automatic differentiation, where we relied on CasADi [49].

Because we fix the number of mesh intervals for every simulation problem, we made the mesh interval length a variable to accommodate for different possible stride lengths at a given speed.

## Results

### Optimizing model body-segment dimensions improves sprinting (17%) and marathon performance (36%)

Optimizing the model's body-segment dimensions for sprinting increased maximum speed by 17% (Fig 2A; 9.49 m/s for the sprint-optimized body-segment dimensions compared to 8.13 m/s the generic model). Optimizing body-segment dimensions for marathon running reduced the lower limb energy cost of running a marathon at 3.3 m/s by 36% (Fig 2B; 2074 kcal for the marathon-optimized body-segment dimensions compared to 3267 kcal for the generic model). The sprint-optimized model was heavier than the generic model, but height did not change markedly (Fig 2C and 2D; 83.6 kg, 1.81 m for the sprint-optimized model compared to 75.2 kg, 1.81 m for the generic model), while the marathon-optimized model was lighter and shorter than the generic model (Fig 2C and 2D; 54.2 kg, 1.76 m for the marathon-optimized model). Predictions for mass and height fell within the range of values found in the top seven fastest all-time male 100m sprinters and marathon runners (Fig 2C and 2D).

Analysis of the optimized body-segment dimensions and resulting joint-torque capacities revealed the hip muscles as important drivers of sprinting and marathon running performance (Fig 2E and 2F). The increased pelvis depth in both the sprint-optimized and marathon-

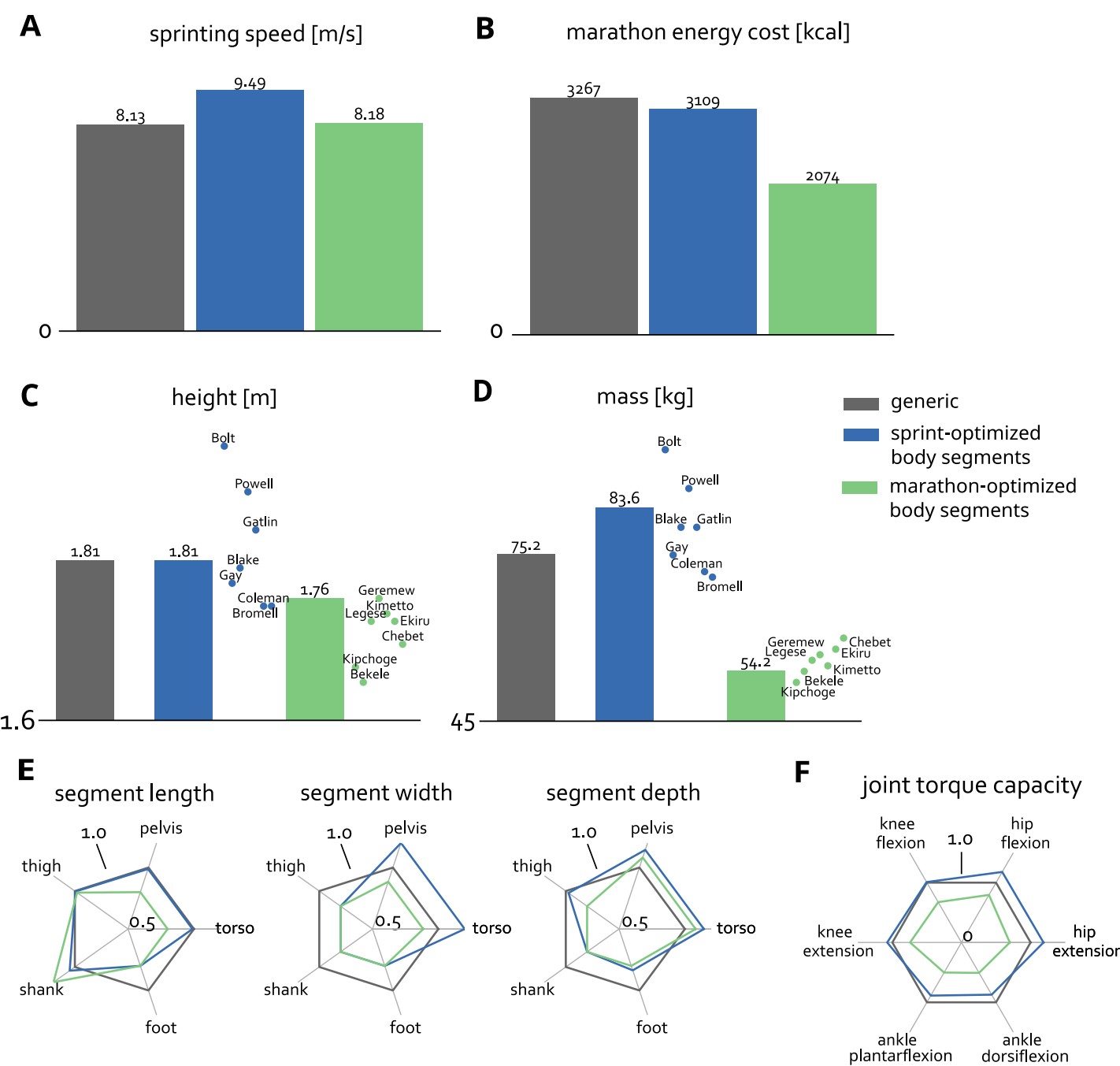

**Fig 2.** Sprinting speed (A), marathon lower limb energy cost (B), height (C), mass (D) for the generic model and models with optimal body-segment dimensions for sprinting and marathon running. The dots and names in (C) and (D) represent the seven all-time fastest male 100m and marathon runners. In (E) the scaling of individual body segments are shown for the three dimensions: length, width, and depth. The scaling factors are normalized to the generic model. In (F) the figure shows the joint torque capacity of the sagittal plane degrees of freedom for the different models normalized by the capacity of the generic model.

optimized models (Fig 2E) increased the moment arms of the hip flexors and extensors, leading to increased hip flexion and extension joint torque capacity (Fig 2F). In the sprint-optimized model, the stronger muscles, due to increased mass, further contributed to increased joint-torque capacity. Despite the smaller and weaker muscles in the marathon-optimized

model, which were reduced to between 55–75% of the generic model because the model was shorter and thinner, the hip flexion and extension capacities were maintained at about 80% and 72%, respectively, of the generic model due to increased moment arms. At the knee and ankle, the sprint-optimized model showed little change in torque capacity, while the marathon-optimized model showed reduced capacity at both joints, due to lower mass and smaller muscles.

A longer shank was beneficial for both sprinting (+4%) and marathon running (+20%), whereas thigh lengths were maintained in the sprint-optimized model and slightly shorter (1%) in the marathon-optimized model compared to the generic model (Fig 2E). Width and depth of the shank, thigh and foot segments were reduced in both the sprint- and marathon-optimized models, reducing lower-limb inertia.

The sprint-optimized and generic models sprinted with similar step lengths (1.92 m and 1.93 m, respectively), but the sprint-optimized model sprinted with increased step frequency (4.9 Hz vs 4.2 Hz). The marathon-optimized and generic models both ran at marathon pace with a 0.94 m step length and step frequency of 3.5Hz.

## Training the hip muscles and plantarflexors is beneficial for sprinting

The model "trained" to optimize sprint performance realized a 9.2% higher sprinting speed (8.88 m/s) compared to the generic model (8.13 m/s) (Fig 3A), while the model "trained" for the marathon reduced the energetic cost of running a marathon by 1.5% compared to the generic model (Fig 3B). Optimized strength training for marathon running was also beneficial for sprinting, evidenced by a 2.3% increase in sprinting speed for marathon-optimized strength. Marathon performance after optimal sprint strength training changed minimally (0.3%). The sprint-optimized model sprinted with a longer step length than the generic model (2.07 m vs. 1.94 m) and a slightly increased step frequency (4.3 Hz vs. 4.2 Hz). The marathon-optimized and generic model both ran at marathon pace with a 0.94 m step length and step frequency of 3.5Hz.

The model with sprint-optimized muscle volumes had increased joint torque capacity primarily for hip flexion, hip extension, knee flexion, and ankle plantarflexion (Fig 3C). Among the muscles with hip flexion moment arms, iliacus, adductor longus, psoas, and tensor fascia latae were beneficial to strengthen (Fig 3D, hip flexion). Among the muscles with hip extension moment arms, biceps femoris longhead, adductor longus, and semimembranosus (Fig 3D, hip extension) were selected as muscles to strengthen. For the ankle plantarflexors, the optimizer suggested strengthening the soleus, and medial and lateral gastrocnemius (Fig 3D, ankle plantarflexion).

We analyzed joint torques during sprinting to understand how the optimized models capitalized on increased muscle volumes. The temporal profiles of the joint torques for the generic model and the two optimized models are similar (Fig 4). However, the model with added muscle volume optimized for sprinting produced a greater peak ankle plantarflexion torque during stance (Fig 4, ankle torque) and a greater peak knee extension torque (Fig 4, knee torque) compared to the generic model. Shortly after take-off, the two optimized models generated more hip flexion torque (Fig 4, hip torque). In terminal swing phase, the sprint optimized model generated more hip extension torque and both optimized models generated higher knee flexion torque (Fig 4, hip and knee torque).

## Simulator performance and validity

Simulations in this study took 30 minutes to 4 hours to converge, when running on a laptop (11th Gen Intel Core i9 2.5GHz CPU). Prior to these adaptations, an optimization of the type

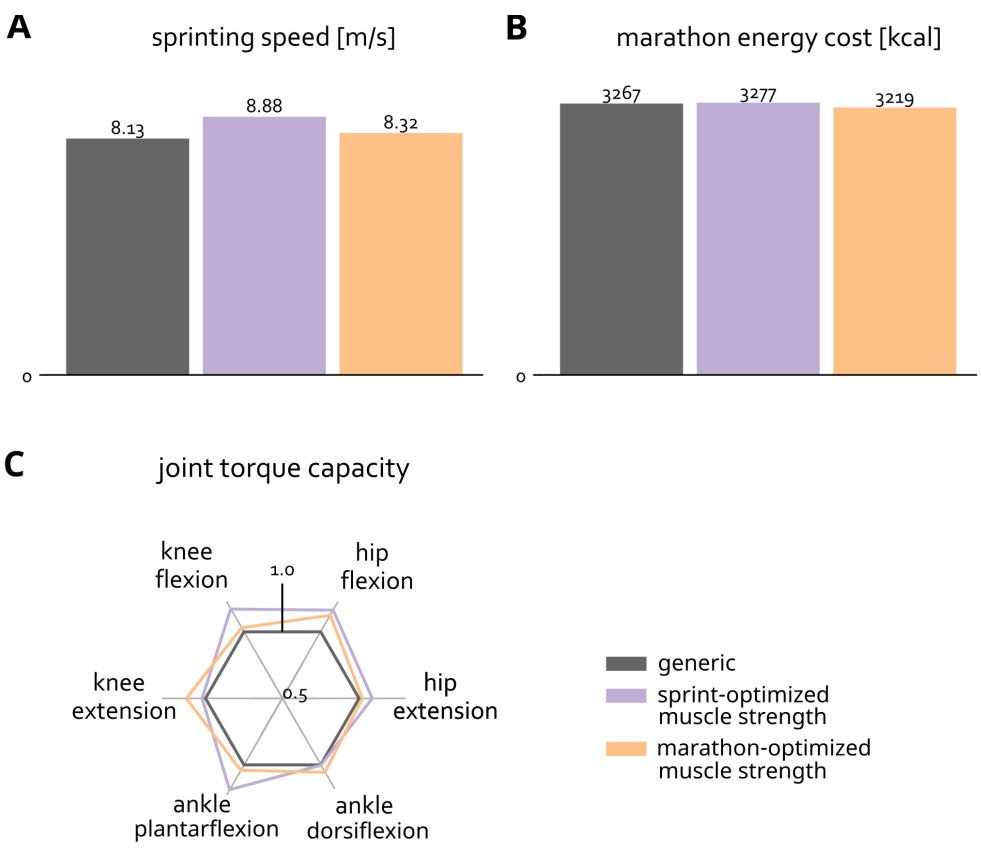

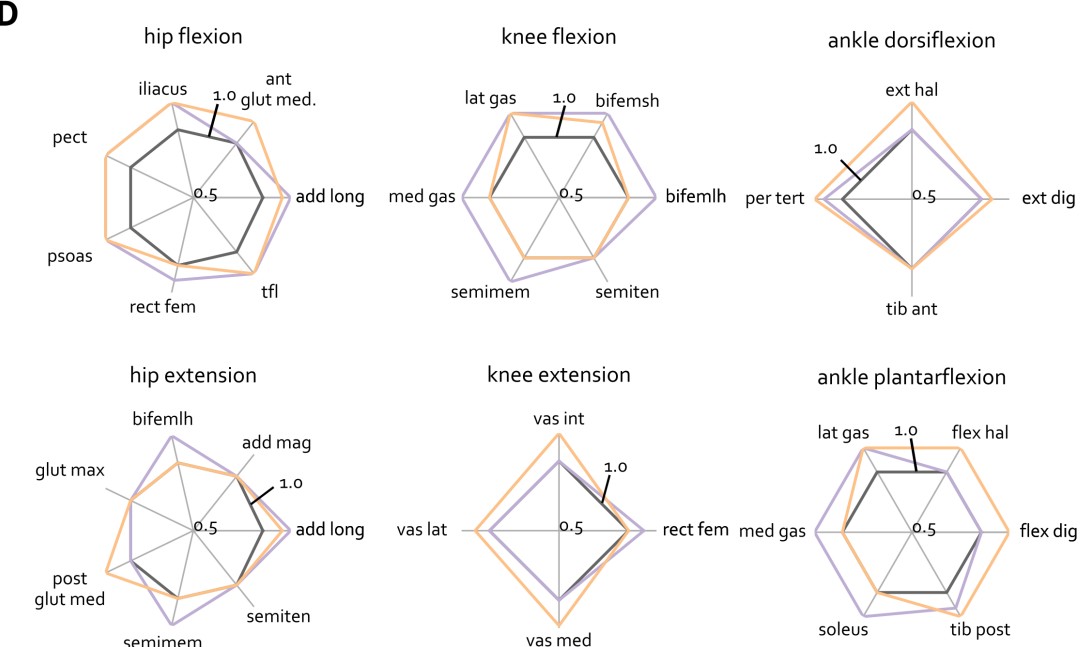

**Fig 3.** Sprinting speed (A) and marathon lower limb energy cost (B) for the generic model and models after optimal 'strength training' for sprinting and marathon running. (C) shows the joint torque capacity of the sagittal degrees of freedom for the different models normalized by the capacity of the generic model. (D) shows normalized muscle maximal isometric force for the different models organized per degree of freedom.

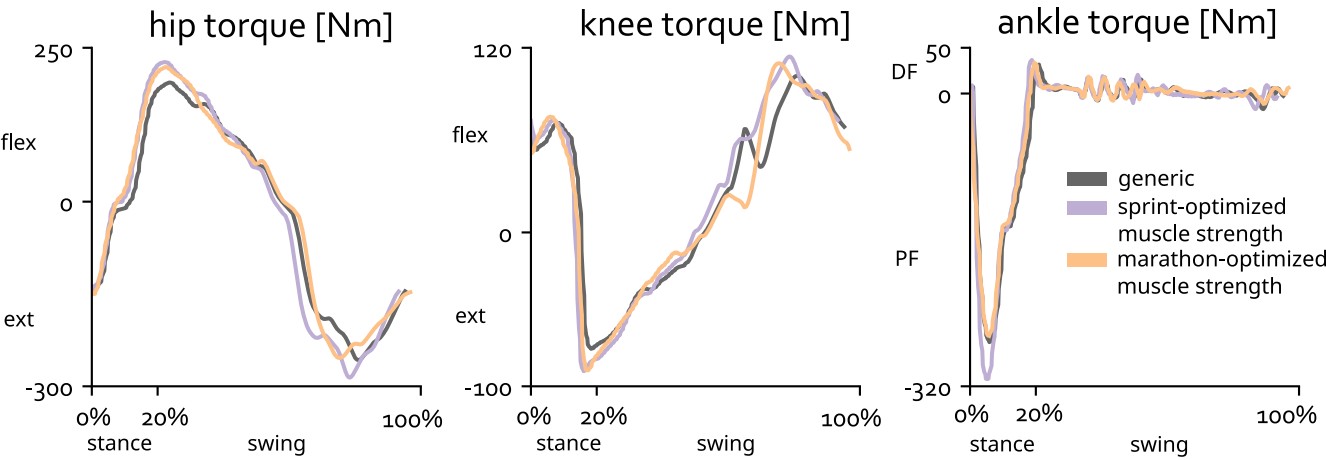

**Fig 4. Joint torques for the sagittal-plane degrees of freedom across the sprinting cycle for the right leg.** FLEX indicates flexion moment, EXT indicates extension moment, DF indicates dorsiflexion moment, PF indicates plantarflexion moment.

used in this study took several days. The range of computation times across several similar optimization problems is expected due to the non-convexity and high non-linearity of the optimization problems [50] that stem from muscle-tendon dynamics [51] and contact dynamics [41,52].

Our simulated kinematics represent many of the key features of experimentally measured kinematics during running and sprinting, giving confidence in the validity of the musculoskeletal model to address our study aims. However, there are also some discrepancies: during swing, simulated hip and knee flexion angles were smaller than observed in experiments for both sprinting and marathon running. A detailed comparison is made in the Supplementary Materials (Figs A and B in S1 Text for marathon running and Figs C and D in S1 Text for sprinting).

Predicted marathon energy consumption for the generic model (3267kcal at 75.2kg) was similar to experimental values measured in 10 recreational marathon runners (2792±235kcal at 72.3kg) [53]. Predicted maximal sprinting speeds were 33% slower than elite sprinting speeds.

## Discussion

Our simulations revealed skeletal geometries and muscle volume distributions that are favorable for sprinting and marathon running, as well as new cause-effect relationships between skeletal geometry, strength, sprinting speed, and marathon running economy. In agreement with experiments [1,19,20,24], sprinting speed improved with additional muscle volume, mainly concentrated at the hip, and marathon economy improved with smaller and lighter body segments and an increased leg-length-to-height ratio. Our simulations also revealed potential mechanisms underlying these relationships. For example, we found that increased hip muscle volume allowed for a faster swing-leg recovery during sprinting, enabling greater step frequency and longer stride lengths. This was the case in our optimizations for both body segment dimensions and muscle volumes. Increased ankle plantarflexor strength was required to generate higher ground reaction forces at these faster sprinting speeds. We also found that an increased shank-to-thigh length ratio and a longer total leg length, without concomitant changes in step length, improved marathon running economy.

Our musculoskeletal simulator is novel since it is differentiable with respect to body-segment dimensions and the inertial properties of a model. We achieved this by (1) formulating the skeleton dynamics to be differentiable with respect to the geometries and inertial properties of the bodies and (2) approximating the computation of muscle wrapping with neural networks, using the musculoskeletal geometry of OpenSim [11] to train the networks. These adaptations allowed us to generate three-dimensional simulations with detailed musculoskeletal models in hours instead of days.

## Musculoskeletal simulation reveals determinants of sprinting performance

Our simulations indicate that increasing strength could improve sprinting performance. In the case of the model with segment dimensions optimized for sprinting, the increase in performance stemmed from both increases in muscle volume (strength) and optimized geometry. This supports and explains the observation that sprinters, and athletes in sports where sprinting gives an important advantage, are muscular and have an increased BMI [17,19,23].

In support of past correlations observed in experimental and simulation studies [19,37], we identified hip musculature as an important limitation for top sprinting speed. Our sprint-optimized models increased hip flexion and hip extension torque capacities by increasing the moment arms and volumes for these muscles. Our simulated sprinting torques show that faster sprinting goes along with higher hip flexion torques during early swing, to bring the swing leg forward, and higher hip extension torques at the end of swing, to prepare for foot contact. Dorn et al. [37] showed that the hip muscles, primarily the gluteus maximus and hamstrings, are key contributors to propulsion at higher running speeds; we build on this work by showing that increasing the moment-generating capacity of these muscles either by increasing moment arms or increasing muscle volumes indeed leads to faster sprinting speeds.

In alignment with an observational study, which showed that knee flexor strength allowed differentiation between untrained age-matched controls, sub-elite, and elite sprinters [19], knee flexion strength was greater in the sprint-optimized models, with concomitant increases in peak knee flexion torques during sprinting. The increased knee flexor capacity may be a byproduct of the optimizer aiming to maximize ankle plantarflexion capacity, thereby increasing the capacity of the gastrocnemii. Our simulations indicate that knee extension strength may not be a key limiter of sprinting performance, which might explain why the study by Miller et al. [19] could not differentiate elite and sub-elite sprinters based on knee extensor muscle volumes. On the contrary, several other experimental studies associate greater maximal knee extensor torques with faster sprinting [54,55]. However, this does not mean increasing knee extensor strength is required to run faster. In our simulations, both models optimized for sprinting produced greater knee extension torques at midstance, but they did not require a significant increase in knee extension joint torque capacity to do so. These higher torques with constant muscle force generating capacities could be explained by the interplay of several factors including higher muscle activation levels, different kinematics resulting in larger moment arms, and the muscles operating at a more favorable section of the force-length and force-velocity curves.

Our sprint-optimized simulations showed that increasing ankle plantarflexion capacity improved sprinting performance, but prior experimental studies are conflicting. Several studies could not correlate ankle muscle volume to sprinting performance [19,56], but others associated increased ankle plantarflexion strength with increased sprinting performance [30,54]. We found that the model with marathon-optimized body segments slightly increased maximum sprinting speed while having reduced absolute plantarflexion muscle volume, which may help explain the conflicting results in prior experiments.

Two mechanisms have been proposed in previous work as limiters of maximal running speed: (1) decreasing ability to support body weight due to decreased stance time with increasing running speed [57,58] and (2) inability to quickly propel the limb forward during early swing, followed by energy removal in late swing to arrest forward motion and accurately position the foot [59]. Miller et al. [34] analyzed the effect of changing the force-velocity relationship on maximal running speed and analyzed whether changes in the force-velocity relationship affect running speed through these mechanisms. While Miller et al. found that both mechanisms could play a role, our simulations do not provide strong evidence for mechanism (1). Stance-phase hip extension torques did not increase at higher running speeds and although stance knee extension torques increased this did not require stronger vastii. The increases in hip flexion and extension torques at higher speeds along with stronger hip and knee flexor and hip extensor muscles in our optimizations provide support for mechanism (2).

In agreement with a study by Tomita et al. [18], we found that a long shank and high shank-to-thigh ratio was optimal for sprinting. Because the shank is lighter than the thigh, such geometry decreases lower limb inertia with respect to the hip, increasing the hip flexion acceleration for a given level of hip flexion torque during early swing, without sacrificing leg length. This advantage affects sprint technique, with the faster sprint-optimized model showing increased step frequency rather than longer step length.

## Lower body mass and specialized skeletal geometry affects marathon performance more than increasing strength

When optimizing skeletal geometries, we constrained the body mass index (BMI) to be within a healthy range (between 17.5 and 25.5). The lower limit captures the fact that athletes, and nearly all humans, should maintain a BMI above 17.5 to prevent low bone density [60,61], hormonal issues [62], low energy availability [63], and an array of other health risks [64]. Because our musculoskeletal model and optimization framework does not capture the detrimental performance and health effects of these phenomena, we introduced this lower limit. It is important to note that the optimization did not converge to a minimal achievable mass and predicted a height and mass similar to top marathon runners [1,24]. Mass could have been further reduced within the imposed BMI constraint by lowering shank and thigh length, which would have diminished performance.

Our simulation results indicate that longer legs, a high leg-length-to-height ratio and a high shank-to-thigh length ratio improve marathon economy, in line with experimental findings [25,27]. Despite having longer legs, the model with segment dimensions optimized for marathon running maintained stride length compared to the generic model when running at our prescribed marathon pace. This seems counterintuitive, but experimental research has shown that optimal stride length does not correlate well with leg length [65] and is subject specific [66]. Further, stride length varies with speed, requiring different propulsive forces [67]. Reduced muscle strength might explain a smaller optimal stride-to-leg-length ratio for the marathon-optimized model compared to the generic model, since larger steps lead to increased peak joint torques and muscle forces.

The small effects of strength training on marathon performance (0.3% improvement) indicate that strength training is less important for marathon runners than for sprinters. Therefore, we do not discuss the redistribution of muscle volume for this case in detail, and suggest that the results for the marathon-optimized model in Fig 2D should be interpreted with care. Nevertheless, the marathon-optimized body segment dimensions indicate hip musculature as a potential limiter, as hip flexion and extension capacity was mostly maintained in this model in spite of the lower overall muscle volume. Strength training also has potential benefits for injury

prevention [68] and could give a competitive edge when a race comes down to a sprint at the end.

Simulated marathon running had a step frequency (210 steps per minute) that is higher than what is typically observed in real-world sprinting (170 steps per minute) [69]. At a specific speed, a runners' preferred step frequency coincides with the metabolic energy minimum [69], and model errors that affect metabolic energy computation could explain the higher simulated step frequency. For example, the metabolic energy model used in our study might underestimate the increasing metabolic cost of producing (the same) cyclic force at increased rates. This phenomenon is described by Doke and Kuo [70] and also suggested by Swinnen et al. [71]. It is hypothesized to be associated with activation-deactivation dynamics, and the cost of calcium transport into the sarcoplasmic reticulum (SR) [72]. Next, the presence of an activation squared in our objective function might bias the model to run with higher cadence as it has been shown that some muscles operate at lower average activations at higher cadences [69].

## Limitations

The results of our simulations are not guaranteed to be applicable to each individual because the specific changes to create an optimized model depend on the body segment dimensions and muscle strengths of the generic model. The body segment geometries, inertial properties, and muscle geometry of the generic model are the synthesis of a series of carefully executed studies [44,73–75] that resulted in a model to represent an average male of 1.81m and 75.1kg. The muscle volumes of individual muscles in the model are based on detailed measurements of muscles in cadavers of both young [76] and older adults [77]. The specific tension of individual muscles, which scale the muscle volumes to maximal isometric forces, were scaled to match maximal torque-angle relationships established by dynamometer measurements and enable the model to reach realistic vertical jumping heights [44,78]. Since the development of this model, many studies have proven its usefulness for simulations of different tasks. There are many ways that a musculoskeletal model can be personalized to represent different individuals. For example, the researchers in past studies have scaled a generic OpenSim [36] model's muscle volumes to simulate elite sprinting [38] and adapted musculoskeletal geometry for deep squats [79]. Additional muscle parameters that could be individualized and potentially contribute to performance in runners are the force-length, force-velocity and muscle activation times. These parameters vary according to muscle fiber type distribution, which is specialized in sprinters and distance runners [42,43]. In the current study, we chose to keep fiber type constant to isolate the effects of musculoskeletal geometry and muscle mass distribution. Thus, while the generic model has been carefully developed and tested, the specific numerical results presented here depend on its properties. It is possible, however, to overcome this limitation and to understand how performance depends on musculoskeletal parameters for a specific person if a personalized musculoskeletal model is available. Developing personalized models is a challenge for future research.

Another limitation of our study, and an interesting avenue for future work, lies in how we scaled muscle volumes with scaling of the skeleton segments. A larger person will have greater muscle volume than a smaller person with the same fat percentage. However, the assumption used in this study of a linear increase in muscles volume distributed uniformly across muscles influences simulated results. For example, in our simulator, the strength of all muscles increased when the optimizer increased the size of the torso or upper arm. An alternate approach to explore in future work is to assign each muscle to one or more segments and scale muscle volume with the volume changes of the assigned segments.

Our simulations of sprinting have lower hip and knee flexion angles during swing compared to experimental measurements (Fig C in S1 Text), as observed in prior sprinting simulations [80]. This might be due to passive hip and knee extension moments becoming large in the model at the more extreme flexion angles. This is a known limitation of the model we used, and attempts to mitigate this issue have been performed [79]. However, sprinting simulations of the generic model with passive forces disabled did not result in larger knee flexion angles, thus we decided not to decrease the passive muscle forces since this may compromise the simulation of other motions for which the original model was developed.

The speeds at which we simulated marathon running and that we found for sprinting are below elite level. The lower sprint speed is mainly due to the relative weakness of the muscles in our generic model. When multiplying the maximal isometric force of all muscles in the generic model by a factor of 2, as was done in several previous sprinting simulation studies [36,38], the maximal running speed increased to 11.1m/s, which is closer to the fastest ever recorded speed of 12.42m/s. The speed of 3.33m/s was chosen as a representative speed for the marathon because it has a similar ratio to elite marathon running speed as the maximal running speed of the generic model to elite sprinting speed.

Our optimization is limited to maximizing speed and minimizing energy consumption and ignores other important factors. For example, the segment scaling could potentially lead to very narrow and long bone geometries that might be more prone to injury. Also, the optimal running techniques simulated might induce high force peaks that generate painful joint loads, and we did not limit these forces in the simulations. Although the vertical impulse is similar in simulation and experiment, the simulated vertical ground reaction forces have a higher peak compared to experimental data (Fig D in S1 Text).

## Conclusions and future directions

Our open-source simulator was able to simultaneously optimize many musculoskeletal parameters in a three-dimensional simulation of running to uncover determinants of sprint and marathon running performance. The simulator could also be used to optimize performance of other athletic tasks such as jumping and accelerative running, which are important for many sporting events. While the present study focused on athletic performance, our approach could also be used in other applications where muscle strength or musculoskeletal geometry affect mobility. For example, the simulator could determine minimal strength requirements to safely perform activities of daily living, guide strength training interventions in elderly people who struggle with specific activities of daily living, or help plan musculoskeletal surgery that aims to enable or improve the performance of specific activities by changing musculoskeletal geometry.

## Supporting information

**S1 Text. .docx file with comparison of experimental and simulated kinematics and evaluation of muscle-length and moment arm approximation.**
(DOCX)

**S1 Video. A video of the generic model running at marathon pace.**
(MP4)

**S2 Video. A video of the generic model sprinting at maximal velocity.**
(MP4)

**S3 Video. A video of three models running at marathon pace: the generic model and the models with optimized musculoskeletal geometry for sprinting and marathon running.** (MP4)

**S4 Video. A video of three models sprinting at maximal velocity: the generic model and the models with optimized musculoskeletal geometry for sprinting and marathon running.** (MP4)

**S5 Video. A video of three models running at marathon pace: the generic model and the models with optimized muscle volumes for sprinting and marathon running.** (MP4)

**S6 Video. A video of three models sprinting at maximal velocity: the generic model and the models with optimized muscle volumes for sprinting and marathon running.** (MP4)

## Author Contributions

**Conceptualization:** Tom Van Wouwe, Jennifer Hicks, Scott Delp, Karen C. Liu.

**Data curation:** Tom Van Wouwe.

**Formal analysis:** Tom Van Wouwe.

**Funding acquisition:** Tom Van Wouwe, Jennifer Hicks, Scott Delp, Karen C. Liu.

**Investigation:** Tom Van Wouwe.

**Methodology:** Tom Van Wouwe, Karen C. Liu.

**Project administration:** Tom Van Wouwe.

**Resources:** Tom Van Wouwe.

**Software:** Tom Van Wouwe.

**Supervision:** Tom Van Wouwe, Jennifer Hicks, Scott Delp, Karen C. Liu.

**Validation:** Tom Van Wouwe.

**Visualization:** Tom Van Wouwe.

**Writing – original draft:** Tom Van Wouwe, Jennifer Hicks, Scott Delp, Karen C. Liu.

**Writing – review & editing:** Tom Van Wouwe, Jennifer Hicks, Scott Delp, Karen C. Liu.

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
