## [Decision Letter · Decision Letter 0]

25 Sep 2023

Dear Dr Van Wouwe,

Thank you very much for submitting your manuscript "Simulated musculoskeletal optimization for sprinting and marathon running" for consideration at PLOS Computational Biology.

As with all papers reviewed by the journal, your manuscript was reviewed by members of the editorial board and by several independent reviewers. The reviewers overall found the paper to be well written, and use novel methods to address an interesting question. The reviewers did raise a number of concerns however, and made numerous constructive suggestions for how the paper might be further improved. In light of the reviews (below this email), we would like to invite the resubmission of a significantly-revised version that takes into account the reviewers' comments.

We cannot make any decision about publication until we have seen the revised manuscript and your response to the reviewers' comments. Your revised manuscript is also likely to be sent to reviewers for further evaluation.

Sincerely,

Adrian M Haith

Academic Editor

PLOS Computational Biology

Stacey Finley

Section Editor

PLOS Computational Biology

Reviewer's Responses to Questions

**Comments to the Authors:**

Reviewer #1: Please see attached review.

Reviewer #2: The authors of this manuscript have developed a differentiable musculoskeletal simulator and used it to optimize body segment dimensions and muscle volumes for sprinting and marathon running. The technical contributions of this study are excellent but there are numerous weaknesses with the application to running. In both the technical and scientific aspects there are also numerous issues with the manuscript that impair clarity and negatively impact the potential of this paper in its current form. In my review below I have provided specific suggestions in an effort to help the authors address these concerns.

General Comments:

This manuscript does not provide page or line numbers, which makes it very cumbersome for the reviewer to provide specific comments on the paper. I strongly urge the authors to include page and line numbers in their future journal submissions. In places below where I refer to a page number I started counting from the page with the title, authors, and abstract as page 1.

The manuscript is replete with the word “higher” when there is no intent to refer to vertical position. This sometimes impairs the clarity of passages, and it can lead to some awkward phrases, such as “a higher relative lower” (Page 4). In most, if not all, of these cases it would be preferable to use a word like “greater” in place of “higher.”

The manuscript ignores some of the key literature that tries to determine factors affecting peak sprinting speed (e.g., page 11: Determinants of sprinting performance.) Addressing this literature would add depth and substance to the discussion section. See for example: Chapman & Caldwell, Journal of Biomechanics, 1983, Weyand et al., Journal of Applied Physiology, 2010, Miller et al., Journal of Biomechanics, 2012. The authors have also missed arguably the most closely related study to their own, in which muscle volume is optimized to maximize vertical jump height. (Wong et al., PLoS One. 2016). (e.g., relevant to the bottom of page 4, and possibly in the discussion section). Addressing these issues and other mentioned below would enhance the scientific merit of what is otherwise a sound study.

Specific Comments:

Normally I favor brief titles. However, similar wording to the current title is often used to describe optimizing only the motion of a musculoskeletal model. By adding a few more words the authors could make it clear that they also optimized the structure of the musculoskeletal system, which might not be obvious to many potential readers when the first see the current title.

Page 1: “However, our ability to understand how these parameters affect task performance has been limited due to the high computational cost of modelling the necessary complexity of the musculoskeletal system and solving the requisite multi-dimensional optimization problem.”

Comment: The wording could be improved because it assumes right away that the only way to study these matters is using a modeling approach.

Page 1: “Our simulation results can help sprint and marathon runners customize strength training”

Comment: This statement overreaches. No runner is going to read this one study and know how to change their training. This study can at best suggest things that can be tested experimentally for verification before being used by athletes and coaches.

Page 3: “Falisse et al. [8] enabled automatic differentiation, a faster alternative to finite differentiation, for musculoskeletal simulation…”

Comment: It is fine to mention the Falisse study here due to the relevance of the specific approach used in the present study. However, it comes across as if that study might have been the first to avoid the use of slow finite difference. In fact, there is a long history of people using collocation methods with exact derivatives to enable rapid musculoskeletal simulation, and this should at least be acknowledged in that passage. For example see Kaplan and Heegaard (J Biomechanics, 2001) and van den Bogert et al. (Procedia IUTAM, 2011).

Page 3: “To exercise this new simulator…”

Comment: Since this is a study about exercise (running) I suggest using a different word here to avoid confusion.

Page 4: “stronger hip muscles and a deeper pelvis”

Comment: Please clarify for the reader what a “deeper” pelvis means.

Page 4: Last paragraph that continues onto the top of page 5.

Comment: Given how little there is that addresses the running application in this paper, it may be worth mentioning the AnyBodyRun tool (https://www.anybodyrun.com) as it shares some of the same goals as this paper, while differing in many details. I am not suggesting AnyBodyRun is of comparable complexity to the present work, just that there is some overlap.

Pages 5-10: Results section.

Comment: There are long passages in the Results section describing the methods that were used. This makes the Results section much longer than it needs to be and makes the reader have to work to identify the findings among the descriptions of the methods. Please keep the methods in the Methods section. With the “methods at the end” structure of the manuscript if there is a need to cue the reader in certain places about the approach, keep it as brief as possible (e.g., a sentence or phrase).

Comment: It might be helpful for the readers to include a results figure that shows the optimally scaled models to help visualize overall difference in body dimensions, and possibly zoomed in to see effects on moment arms (though perhaps the latter is not possible with this approach).

Page 6: “Simulated kinematics and ground reaction forces for running at 3.33m/s and sprinting with the generic model were similar to experimental values”

Comment: The statement “were similar to experimental values” is disingenuous. This one short phrase does not align with the more nuanced comparisons described in the Supplement, and even then, the results are not all similar. For example, the simulated pelvis tilt (different magnitude) and pelvis rotation (opposite phasing) are not similar to the experimental data.

Page 6: “2074 kcal for the marathon-optimized body-segment dimensions compared to 3267 kcal for the generic model”

Comment: I think these energy consumption values are in the right ballpark for the marathon. It would be helpful to provide some values from the literature for reference so the reader can understand if these predictions are accurate or not.

Page 8: “The sprint-optimized and generic models sprinted with similar step lengths (1.92 m and 1.93 m, respectively), but the sprint-optimized model sprinted with higher step frequency (4.9 Hz vs 4.2 Hz). The marathon-optimized and generic models both ran at marathon pace with a 0.94 m step length and step frequency of 3.5Hz.”

Comment: Much as with the preceding energy consumption values, it would be helpful to provide a range of experimental values from the literature to help the reader understand how accurate these stride parameter predictions are.

Page 11 and 13: Marathon running economy

Comment: The present study can establish cause-and-effect in the sense that something is changed in the model and the effects on running performance can be readily determined. However, it is worth being a bit more circumspect as to how much this study reveals the “mechanisms” behind these effects. As one specific example, on page 11 it is stated that shank-to-thigh ratio, leg length and step length influence running economy, but then on page 13 when this is discussed there is the same level of informed speculation as one would expect in an observational study. There is a balance to be found regarding what is, and what is not, explained in this study.

Page 12: “both models optimized for sprinting produced greater knee extension torques at midstance, but they did not require a significant increase in knee extension joint torque capacity to do so.”

Comment: This is another example where the “determinants” of sprint performance feels to have a bit been oversold. This is certainly an interesting finding, but can you explain *how* is greater knee extension torques being produced without a greater capacity to do so?

Page 15: “For example, the simulator could determine minimal strength requirements to safely perform activities of daily living, could guide strength training interventions in elderly people, or estimate the effects of musculoskeletal surgery.”

Comment: It is a bit of a leap to go from identifying the largely (if not entirely) genetically determined factors that favor sprint and marathon running performance to ergonomic applications or predicting surgical outcomes. It might be helpful to take a few sentences to better explain future applications rather than try to cover it all within the final sentence of the Discussion section.

Page 15: “The third and fourth simulations”

Comment: After this statement the authors stop counting out the simulations for the reader. It would be helpful to continue to do so throughout. I also suggest moving the table that is currently in the supplement to the main document to help the reader understand and keep track of the various simulations.

Page 15: “muscle insertion points”

Comment: Do the authors mean “muscle attachment points”? (“insertion” refers only to the distal attachment).

Page 15: “Importantly, when scaling the skeleton, we chose to scale muscle volumes proportionally to the change in whole body mass and adjust body mass when scaling the body segments by assuming constant density. As such, a heavier model has stronger muscles.”

Comment: Please provide more detail on the scaling approach. This is a critical part of the study and it was difficult to fully understand from this brief statement.

Page 16 (Fig 4 caption): “where we turned the state-of-the-art from non-differentiable to differentiable computation”

Comment: Replace “state-of-the-art” with whatever was actually converted.

Page 18: “are bounded between -1 and 1 with 150 as in”

Comment: Should this be “150 N m”?

Page 18: “the talus, calcaneus, toes, shank, thigh, upper arms, lower arms, hands”

Comment: Consider replacing “calcaneus“ with something like “rear- and mid-foot” or explain that “calcaneus” does not actually mean just the calcaneus bone (this is due to an unfortunate naming scheme in some OpenSim models, similar to calling the shank segment the “tibia”).

Page 19: “and as such changing the muscle maximal isometric force”

Comment: While p_V_muscle is applied to muscle volume it is described as changing maximal isometric force. As such, does it only affect the PCSA or does it also affect Lm_opt? Perhaps this could be explained better.

Related comment: On page 23 there is a statement that seems relevant to this but is hard to understand “When simulating strength training, the increase of individual muscle physiological cross-sectional areas was limited to 20% and the total increase in muscle volume summed over all muscles was limited to 5% of the initial total muscle volume.” Please consider revising for clarity.

Page 19: “technical contribution of this work”

Comment: Change the wording so it is clear whether “this” refers to the present study or that of Falisse.

Page 20: “We chose the respective lower and upper limits of 0.8 and 1.2 as these cover most of the variation in an adult population.”

Comment: Please provide the reference that supports this statement.

Page 21: “We put the model in that joint pose and observed the muscle tendon length and moment arm with respect to the joint coordinates that are actuated by the muscle.”

Comment: Given the large number of poses the model was placed in throughout the joint ROM, how did the authors handle the cases where (I assume) there would be some discontinuities in MT lengths and moment arms?

Page 21: “We then trained a separate neural network for each muscle using Adam [66] with a mean squared error loss over 1,000 epochs and a batch size of 64.”

Comment: Please briefly explain what “Adam” is and why it was selected.

Page 21: “With the described set-up we confirmed that our predictive simulations of walking and running at 3.33m/s yielded kinematics and muscle activation very close to the those with the original OpenSim model and for OpenSim models with all scaling factors at 0.85 or 1.15.”

Comment: I am confused by the reference to “our predictive simulations of walking” as there are no walking simulation results presented in this paper, which is about running. Please clarify.

Page 22: “we solved a trajectory optimization problem where either the skeleton scaling parameters (ps) or the scaling of muscle volumes (pVmuscle) were added to the optimization variables.”

Comment: Perhaps I missed it, but was there a reason to not include the case where both the skeleton and muscles were scaled in the same optimization? Was that computational prohibitive, or was there some other reason? If this is not already addressed then adding a brief statement seems appropriate.

Page 22: “with Jmetabolic the muscle metabolic energy based on the Bhargava model [65] of metabolic

energy expenditure”

Comment: In the paper by Falisse the metabolic energy rate was squared (their Eq 2.2). Was that term also squared in the present study?

Page S2 (2nd page of the supplement): “The pelvis and lumbar degrees of freedom have similar waveforms”

Comment: This statement is not true for the pelvis rotation. The simulated data has nearly the opposite phasing compared with the experimental data.

Page S2: Figure S1

Comment: The figure on this page would be clearer if means and SDs bounds were plotted rather than the individual subject lines.

Page S3: “We compared our sprinting simulation to a sprinting gait cycle from an individual running at about 9.45m/s”

Comment: Please also mention the simulation running speed here.

Reference section

Comment: The reference section is not up to typical standards. There are articles that appear more than once, there are references with information missing (e.g., missing journal name), and inconsistent formatting.

**Have the authors made all data and (if applicable) computational code underlying the findings in their manuscript fully available?**

Reviewer #1: Yes

Reviewer #2: Yes

PLOS authors have the option to publish the peer review history of their article (what does this mean?). If published, this will include your full peer review and any attached files.

Reviewer #1: No

Reviewer #2: No
---

## [Decision Letter · Decision Letter 1]

4 Feb 2024

Dear Dr Van Wouwe,

We are pleased to inform you that your manuscript 'A simulation framework to determine optimal strength training and musculoskeletal geometry for sprinting and distance running' has been provisionally accepted for publication in PLOS Computational Biology.

Best regards,

Adrian M Haith

Academic Editor

PLOS Computational Biology

Stacey Finley

Section Editor

PLOS Computational Biology

Reviewer's Responses to Questions

**Comments to the Authors:**

Reviewer #1: The authors satisfactorily addressed my comments.

Reviewer #2: I appreciate the thoughtful and comprehensive approach the authors have taken in revising their manuscript. This paper will make a valuable contribution to the literature in musculoskeletal modeling and simulation, and it addresses an interesting question in human running performance. I have no further comments.

**Have the authors made all data and (if applicable) computational code underlying the findings in their manuscript fully available?**

Reviewer #1: Yes

Reviewer #2: Yes

PLOS authors have the option to publish the peer review history of their article (what does this mean?). If published, this will include your full peer review and any attached files.

Reviewer #1: No

Reviewer #2: No

---

## [Editor Report · Acceptance letter]

15 Feb 2024

PCOMPBIOL-D-23-01242R1 

A simulation framework to determine optimal strength training and musculoskeletal geometry for sprinting and distance running

Dear Dr Van Wouwe,

I am pleased to inform you that your manuscript has been formally accepted for publication in PLOS Computational Biology. Your manuscript is now with our production department and you will be notified of the publication date in due course.

With kind regards,

Anita Estes
